# Roles of the Crp/Fnr Family Regulator ArcR in the Hemolysis and Biofilm of *Staphylococcus aureus*

**DOI:** 10.3390/microorganisms11071656

**Published:** 2023-06-25

**Authors:** Tongtong Fu, Zheng Fan, Yujie Li, Zhoufei Li, Hanqing Zhao, Yanling Feng, Guanhua Xue, Jinghua Cui, Chao Yan, Lin Gan, Junxia Feng, Jing Yuan, Fuping You

**Affiliations:** 1School of Basic Medical Sciences, Peking University, Beijing 100020, China; 2Department of Bacteriology, Capital Institute of Pediatrics, Beijing 100020, China; 3Department of Life Science and Medicine, University of Science and Technology of China, Hefei 230026, China

**Keywords:** *Staphylococcus aureus*, ArcR, biofilm, PIA, hemolysis, Crp/Fnr

## Abstract

*Staphylococcus aureus* is an opportunistic human pathogen that is often involved in severe infections such as pneumonia and sepsis in which bacterial virulence factors play a key role. Infections caused by *S. aureus* are often difficult to eradicate, particularly when they are associated with biofilm. The physiological roles of the Crp/Fnr family regulator ArcR are elusive in *S. aureus*. In this study, it was found that the deletion of *arcR* increased the hemolytic ability and biofilm formation in *S. aureus*. Differential gene expression analysis by RNA-seq and real-time quantitative reverse transcription PCR showed that genes associated with hemolytic ability (*hla* and *hlb*) and biofilm formation (*icaA, icaB, icaC* and *icaD*) were significantly upregulated compared with those in the wild-type strain. The results revealed that ArcR regulated the expression of the *hla* and *ica* operon by binding to their promoter regions, respectively. This study provided new insights into the functional importance of ArcR in regulating the virulence and biofilm of *S. aureus.*

## 1. Introduction

*Staphylococcus aureus* is a globally prevalent pathogen that causes a variety of infections ranging from slight infections of the skin to serious cephalomeningitis [1,2,3]. *S. aureus* produces many virulence factors such as exotoxins, enzymes, and surface protein adhesins contributing to the initiation of adherence, the inhibition or avoidance of the host immune system, and the survival and expansion of *S. aureus* [4,5]. Among these virulence factors, α- and β-hemolysins, as membrane-damaging toxins, are pore-forming toxin (PFT) factors which are encoded by *hla* and *hlb*. α-Hemolysin plays an important role in acute *S. aureus* infections, including skin and bloodstream infections. [6]. Multiple regulators participate in the regulation of *hla* in *S. aureus*. The two-component system (TCS) of SaeRS activated *hla* transcription via a well-defined SaeR binding site [7]. RNAIII performed as an antisense RNA to active *hla* mRNA translation [8]. Agr and SarA also affected the expression of *hla* in vitro and in vivo [9].

Biofilm formation increases the antibiotic resistance and host defense of bacteria contributing to persistent and chronic infections [10,11]. In *S. aureus*, polysaccharide intercellular adhesin (PIA) is vital in maintaining the structural integrity of biofilms [12,13,14,15]. PIA is produced by enzymes are that encoded by the *icaADBC* operon, which is responsible for the synthesis, modification, and export of PIA. The *icaR* gene located downstream of the *icaADBC* operon in *S. aureus* encodes a transcriptional repressor that regulates the expression of the *ica* operon [16,17]. In addition, studies have shown that LuxS/AI-2, SarA, MazEF, SrrAB, and YycFG participate in the regulation of the *ica* operon in *S. aureus* [1,18,19,20,21,22,23].

Crp/Fnr family regulators are among the most widely studied transcriptional regulators, and they can respond to a variety of environmental stimuli by activating and repressing the expression of various functional genes across bacteria [24,25]. Previous studies have shown that some Crp/Fnr family members participated in the regulation of virulence and biofilm formation. The Crp/Fnr family member PrfA in *Listeria monocytogenes* and the Cmr and CRPMt in *Mycobacterium tuberculosis* regulated the bacterial virulence-associated genes [26,27,28]. The Crp/Fnr family member PgRsp in *Porphyromonas gingivalis,* the Bcam1349 in *Burkholderia cenocepacia,* and the Lmo0753 in *L. monocytogenes* affected the biofilm production [29,30,31].

It is worth noting that the Crp/Fnr family regulator ArcR shares low overall similarity with other family members. The ORF of ArcR in *S. aureus* shares 45% similarity and overall 19% identity with the ArcR of *Bacillus licheniformis* and is much less similar to the Crp of *Escherichia coli*. Moreover, 18 ArcR shared remarkably low overall similarity through multiple alignments [32]. The low similarity suggests that they differ in the response to environmental stress, so it is necessary to study the function of ArcR in *S. aureus* [32]. ArcR affected the utilization of arginine under anaerobic conditions through the regulation of the pathway genes *arcABDC* [32]. Our previous study showed that ArcR contributed to the tolerance of fluoroquinolone antibiotics in *S. aureus* [33]. In this study, we showed the function of ArcR in virulence and biofilm formation. RNA-Seq and qRT-PCR assays indicated that *icaADBC*, *hla,* and *hlb* were upregulated in the Δ*arcR* mutant compared with the wild type (WT). The β-Galactosidase activity assay provides more direct evidence of the role of ArcR in the regulation of genes in *S. aureus*. The results of the β-Galactosidase activity assay showed that the deletion of *arcR* increased the promoter activity of *hla* and *hlb*. The construction of the Δ*arcR* Δ*icaA* double mutant strain and the detection of PIA production indicated that arcR increased the biofilm formation through *ica* operon. Further study demonstrated that ArcR could bind to the promoter regions of *hla* and *ica* operon promoter regions and regulate their expression.

## 2. Materials and Methods

### 2.1. Bacterial Strains, Plasmids, Primers, and Growth Conditions

The bacterial strains and plasmids are shown in Table 1. The primers used in this study were designed using Primer Premier 5 and are listed in Appendix A. *S. aureus* RN4220 was used to modify the plasmids before transformation into the other *S. aureus* strains. Luria-Bertani (LB) medium was used to culture *Escherichia coli,* and tryptic soy broth medium (TSB) containing 0.5% glucose was used to culture *S. aureus*. Antibiotics were used for constructed plasmid selection and maintenance at the following concentrations: 50 μg/mL kanamycin, 100 μg/mL ampicillin, 15 μg/mL chloramphenicol, and 5 μg/mL erythromycin.

### 2.2. Construction of the S. aureus Mutant and Complementary Strains

The Δ*arcR* mutant and Δ*arcR* Δ*icaA* double mutant strains were obtained as described previously [38]. The Δ*arcR* Δ*icaA* double mutant strain was constructed on the background of the Δ*arcR* mutant. Briefly, the fragments of upstream and downstream were amplified from the NCTC8325 genome and then the fragments were ligated via overlap PCR. The fusion segments were cloned into the temperature-sensitive shuttle plasmid T1-pBTS. The constructed plasmid was transformed into RN4220 and subsequently transformed into the NCTC8325 wild type or Δ*arcR* mutant. The mutant strains were screened using a previously described method and verified via PCR and sequencing [39]. The shuttle vector pCN51 was used to construct complementary strains.

### 2.3. Detection of Biofilm Production

The tube method described previously was used to detect biofilm production under static conditions [40]. The bacteria were incubated in TSB for 14–16 h at 37 °C and 200 rpm. Cultures were diluted at 1:100 with fresh TSB and 200 μL of diluted suspension was added into each well of the 96-well plates. The plates were cultured at 37 °C for certain time courses. The wells were washed with water three times and dried at 37 °C. The wells were stained with 0.5% crystal violet at room temperature for 20 min and washed again with water. Then, the film around the wall and bottom of the wells was observed to confirm biofilm production. For the semi-quantitative assessment of biofilm formation, 100 µL per well of bleaching solution (methanol: glacial acetic acid: H_2_O (*v*/*v*/*v*) = 4:1:5) was added to each well with shaking for 1 h. Absorbance at 560 nm was then measured using an enzyme-linked immunosorbent assay reader.

### 2.4. Confocal Laser Scanning Microscopy (CLSM) for the Observation of Biofilm Production

Confocal laser scanning microscopy (CLSM) was introduced for observing biofilm production and form. To detect the function of ArcR in biofilm production, CLSM was used to analyze the biofilm formation in Δ*arcR* mutant strains. Overnight cultures were diluted with fresh TSB (2% glucose) at a ratio of 1:100. Then, 1 mL of diluted bacteria was added to 15 mm cell culture dishes and cultured at 37 °C for 24 and 48 h to form an artificial biofilm at the bottom. The biofilm at the bottom of the dish was washed with phosphate-buffered saline (PBS) buffer three times and then fixed with 2.5% glutaraldehyde at 4 °C for 1.5 h. After washing twice with PBS, 400 µL of 50 µg/mL FITC-conA solution was added to stain extracellular polysaccharides in the biofilm at 4 °C for 1 h. PBS was used to wash the biofilm and then the bacteria in the biofilm were stained with 10 µg/mL propidium iodide (PI) at 4 °C for 15 min. After washing with PBS, the dish was dried at room temperature. All confocal images were digitized using the CLSM system. Image J 1.53a was used for the analysis of integrated fluorescent intensity, biofilm production, bacterial density, and other parameters.

### 2.5. Determination of Hemolytic Activity

The WT, Δ*arcR* mutant, and *arcR* complementary strains were cultured in TSB at 37 °C. The optical density 600 (OD600) of overnight cultures was adjusted to 5 and 2 μL of the cultures was spotted onto sheep blood agar (SBA) plates for the hemolytic activity assay on solid medium. The SBA plates were left to dry and incubated at 37 °C, and hemolysis zones were measured after 24 h.

Hemolytic activity was quantitatively evaluated by incubating samples with 3% sheep red blood cells. The WT, Δ*arcR* mutant, and *arcR* complementary strains were grown in TSB and collected after being cultured overnight. Then, 100 μL of bacterial supernatants was mixed with 900 μL of PBS buffer containing 3% sheep red blood cells, and then incubated at 37 °C for 30 min. After centrifugation, the absorption at 543 nm of the supernatant was measured. The ddH_2_O containing 3% sheep red blood cells and the PBS containing 3% sheep red blood cells were used as the positive and negative control, respectively. The proportion of hemolytic activity was calculated relative to that of the positive control, which was regarded as having 100% hemolytic activity.

### 2.6. Total RNA Extraction, cDNA Generation, qRT-PCR, and RNA-seq

Overnight cultures were added to fresh TSB at a 1:100 ratio and then grown to a certain cell density. Indicated cultures were collected by centrifugation. The precipitate was resuspended in 100 µL of 1 × RNase-free TAE and then incubated at 37 °C for 6–8 min. An RNAprep Pure Cell/Bacteria Kit (Tiangen, Beijing, China) was used for RNA isolation. The residual DNA was cleared using Recombinant DNase I at 37 °C for 60 min. For reverse transcription, 1 µg of RNA was used for cDNA synthesis. qRT-PCR was performed with cham Q universal SYBR qRT-PCR master mix (Vazyme, Nanjing, China). 16S rRNA was used as the reference target. Each reaction was performed in triplicate parallel experiments.

To perform RNA-seq, RNA was prepared as described above. The RNA quality was detected by agarose gel electrophoresis. The Bacterial rRNA Removal Kit (Vazyme, Nanjing, China) was used to eliminate the rRNA. The quality of RNA was evaluated using the Agilent 2100 BioAnalyzer (Agilent, Santa Clara, CA, USA) prior to library synthesis. The first strand of cDNA was synthesized using M-MuLV reverse transcriptase, DNA polymerase I, and dNTPs. The purified cDNA was end-repaired and sequencing joints were connected. Then, the beads of AMPure XP were used to filter the cDNA of approximately 370–420 bp. The beads were used to purify PCR products. After the library was constructed, it was initially quantified using Qubit 2.0 The insert size of the library was tested using the Agilent 2100 Bioanalyzer. qRT-PCR was used to accurately quantify the effective concentration of the library to ensure its quality. High-quality libraries were pooled for Illumina sequencing. NEBNext Ultra Directional RNA Library Prep Kit for Illumina (NEB, Houston, TX, USA) was used. DESeq2 was used to identify differentially expressed genes.

### 2.7. Construction of LacZ Reporter Vectors

To construct the reporter vectors pOS*hla* for the detection of *hla* expression, a fragment containing the *hla* promoter region and a region of the first 78 bp of the coding sequence of the *hla* was amplified from the *S. aureus* genome with the primers pOS1-P*hla*F/R. The PCR product was purified and then digested with *Eco*R1 and *Bam*H1 at 37 °C for 1 h. The digested *hla* fragment was cloned into pOS1-*lacZ* to obtain reporter plasmid pOS*hla*. The same method was used to construct the reporter plasmids pOS*hlb,* pOS*sbi,* pOS*vraX,* and pOS*splA* using the primers pOS1-P*hlb*F/R, pOS1-P*sbi*F/R, pOS1-P*vraX*F/R, and pOS1-P*splA*F/R, respectively. These reporter plasmids were transformed into the NCTC8325 WT and Δ*arcR* mutant strains.

### 2.8. β-Galactosidase Activity Assay

The WT and Δ*arcR* mutant strains containing LacZ reporter plasmids were cultured in TSB at 37 °C overnight for the β-galactosidase assay. The cultures were diluted at a ratio of 1:100 in TSB with 15 μg/mL chloromycetin and grown for 4 and 8 h. Collected cultures were resuspended by 100 μL of ABT-LSA buffer (40 mM KH_2_PO_4_, 100 mM NaCl, 0.1% Triton X-100, 60 mM K_2_HPO_4_, 50 μg/mL lysostaphin) and then incubated at 37 °C for 30 min. Next, 100 μL of ABT buffer and 100 μL of 4 mg/mL ONPG (o-nitrophenyl-β-d-galactopyranoside) were added to start the reaction. The samples were incubated at 37 °C until a yellow color appeared, and 1 mL of Na_2_CO_3_ was immediately added to stop the reaction. The absorbance of samples was read at 420 nm and units were calculated using the following formula: units  =  (1000 × OD420)/(T × V × OD600), where T (measured in minutes) was the incubation time and V (in milliliters) was the volume of culture used in the assay.

### 2.9. PIA Detection

Overnight cultures were collected by centrifugation at 12,000× *g* and precipitates were stored in a −80 °C refrigerator. The precipitates were resuspended with 50 µL of EDTA (pH 8.0) and then incubated at 100 °C for 5 min. Subsequently, the samples were centrifugated at 12,000× *g* and 40 µL of the supernatant was collected for subsequent experiments. Then, 10 µL of 2 mg/mL proteinase K was added to the supernatants at 37 °C for 30 min. The extracted PIA was blotted onto a nitrocellulose filter membrane using a dot-blot apparatus as previously described [41]. Then, the membrane was dried at 65 °C and soaked in PBS containing 3% bovine serum albumin and 0.05% Tween-20 for 1 h. Next, the membrane was incubated in a solution containing 0.8 mg/mL wheat germ agglutinin conjugated with biotin (WGA-biotin) (Sigma-Aldrich, St. Louis, MO, USA) at room temperature for 1 h. After washing four times with PBS, the membrane was incubated in a solution with 0.1 µg/mL horseradish peroxidase-conjugated streptavidin (HRP- streptavidin) at room temperature for 15 min. After washing as described above, chemiluminescence detection was undertaken for PIA quantification.

### 2.10. ArcR Protein Expression and Purification

ArcR was expressed by the pET28a plasmid in BL21 and purified using the following procedures. The expression strains were grown in LB containing 50 µg/mL kanamycin at 37 °C for 3 h. Then, 0.5 mM isopropyl β-d-thiogalactopyranoside (IPTG) was added to induce ArcR expression at 16 °C for 14–16 h. The cultures were then lysed in buffer (25 mM HEPES, 200 mM NaCl, 5 mM β-mercaptoethanol, 10% glycerol, pH 7.8). This fusion protein was purified though a nickel nitrogen triacetate agarose solution (Qiagen, Hilden, Germany). Lysis buffer containing 100, 200, or 400 imidazole was used to elute the bound protein.

### 2.11. Electrophoretic Mobility Shift Assay (EMSA)

The *icaA* and *icaR* promoter region DNA fragments were amplified from *S. aureus* NCTC8325 genomic DNA. The DNA fragments were incubated with purified ArcR protein at room temperature for 30 min in incubation buffer (25 mM HEPES, 300 mM NaCl, 1 mM dithiothreitol, 10% glycerol, pH 7.8). The incubation mixtures were added to 5 µL of loading buffer and electrophoresed on an 8% native polyacrylamide gel in 1 × Tris-borate-EDTA (TBE) buffer for 1–2 h. The shift bands were detected by staining with 0.5 μg/mL ethidium bromide (EB) and imaged using a gel imager (Bio-Rad, Hercules, CA, USA).

### 2.12. Statistical Analysis

Statistical analyses were performed using SPSS 20.0 (IBM Corp., Armonk, NY, USA). Data are presented as the means ± the standard deviations (SD) and analyzed using one-way ANOVA. Statistically significant differences were indicated by *p* < 0.05 (*), *p* < 0.01 (**), *p* < 0.001 (***). All experiments were performed in biological triplicates.

## 3. Results

### 3.1. Deletion of arcR Increased the Hemolytic Activity in S. aureus

Qualitative and quantitative hemolytic methods were used to determine whether ArcR modulates the hemolytic activity of *S. aureus* NCTC8325. As shown in Figure 1A, the hemolytic zone of the Δ*arcR* mutant was larger than that of the WT after 24 h of incubation. The quantitative of hemolytic activity was performed with the sheep red blood cells, and then the proportion of hemolytic activity was calculated by measuring the optical density at 543 nm. The Δ*arcR* mutant displayed significantly greater hemolytic activity than that of the WT (Figure 1B,C, Appendix A). Activity levels were restored in the *arcR* complementary strain. These results above demonstrated that ArcR participated in the regulation of hemolysis in *S. aureus*.

### 3.2. Deletion of arcR Increased Biofilm Formation in S. aureus

The effect of ArcR on biofilm formation in *S. aureus* NCTC8325 was evaluated by monitoring the WT, Δ*arcR* mutant, and *arcR* complementary strain using 96-well microtiter plate assays. As shown in Figure 2A, the *arcR* mutation resulted in an increased biofilm formation ability. This alteration could be complemented by the pCN51-*arcR* plasmid. The biofilms of the WT, Δ*arcR* mutant, and *arcR* complementary strains were stained with FITC-ConA and PI and were observed by CLSM. FITC-ConA stained the biofilm to green, and PI stained the bacterial cells to red. We observed an increase in adhered cells on the glass dishes in the Δ*arcR* mutant compared with that of the WT (Figure 2B). In addition, the growth curves for the WT, Δ*arcR* mutant, and the *arcR* complementary strain grown in TSB were determined and the results showed no significant differences among these strains (Figure 2C). These findings indicated that ArcR could repress biofilm formation in *S. aureus*.

### 3.3. Gene Expression Profiling of the ΔarcR Mutant

To understand the mechanism underlying the ArcR-mediated regulation of bacterial biofilm and hemolysis, the global gene expression profiles of the WT and Δ*arcR* mutant were determined via RNA-seq. Differentially expressed genes were identified by adjusted *p* < 0.05 and fold change log2 > 1. The results of these experiments revealed 224 of 2638 gene transcripts that were differentially expressed between the WT and Δ*arcR* mutant. A volcano plot of the RNA-seq results showed 69 downregulated and 155 upregulated genes in the Δ*arcR* mutant (Figure 3A). The expression levels of *ica* operon genes *icaA*, *icaB*, *icaC*, and *icaD* increased by 7.4-, 11.0-, 5.1-, and 3.8-fold over levels in the WT. Moreover, the expression levels of the virulence genes *hla*, *hlb*, *sbi*, *vraX*, and *splA* increased by 11.0-, 5.4-, 4.3-, 3.9-, and 4.5-fold over levels in the WT, respectively (Figure 3B). The downregulated and upregulated genes are listed in Appendix A.

### 3.4. ArcR Regulates the Transcription of HEMOLYTIC Genes

The results presented above demonstrated that ArcR is involved in the regulation of hemolysis in *S. aureus*. To further determine whether the expression of hemolytic genes was altered in the Δ*arcR* mutant, qRT-PCR was performed. The qRT-PCR results showed that the expression levels of *hla* and *hlb* were significantly higher in the Δ*arcR* mutant strain at different growth phases than in the WT strain, and these expression changes could be fully restored by *arcR* complementation (Figure 4A–C). A β-galactosidase assay was performed to confirm the negative regulatory effect of *arcR* on the expression of *hla* and *hlb*. The reporter plasmids pOS*hla* and pOS*hlb* were constructed and transformed into the WT and Δ*arcR* mutant strains. The β-galactosidase activities in the Δ*arcR* mutant strain were significantly higher than those in the WT strain (Figure 4D), consistent with the results of the qRT-PCR. These results suggested that the increase in hemolytic activity in the Δ*arcR* mutant strain was mainly due to the increased expression levels of *hla* and *hlb*.

### 3.5. ArcR Influences Biofilm Formation through ica Operon

To verify the RNA-seq results, the transcript levels of genes of the *ica* operon were measured using a qRT-PCR with RNA extracted from the WT strain and its isogenic Δ*arcR* mutant in planktonic bacteria and biofilm. The transcript levels of *icaA*, *icaB*, *icaC*, and *icaD* were upregulated in the Δ*arcR* mutant not only in planktonic bacteria, but also in biofilm (Figure 5A,B). To determine whether ArcR affects biofilm formation by regulating the expression of the *ica* operon, the *icaA* mutant was constructed from the Δ*arcR* mutant. Biofilm production was evaluated in the WT, Δ*arcR* mutant, and Δ*arcR* Δ*icaA* double mutant and was detected using 96-well microtiter plate assays. Biofilm production was lower in the Δ*arcR* Δ*icaA* double mutant than in the Δ*arcR* mutant (Figure 5C). There were no significant differences in the growth rates between the WT, Δ*arcR* mutant, and Δ*arcR* Δ*icaA* double mutant (Figure 5D).

The genes of the *ica* operon encode proteins involved in PIA synthesis. The production of PIA is important for biofilm formation in *S. aureus*. To investigate whether ArcR inhibits biofilm formation by regulating *ica* operon and subsequent PIA production, we detected the PIA production. As shown in Figure 5E, the PIA production in the Δ*arcR* mutant was higher than that in the WT. Taken together, these results indicated that ArcR inhibited biofilm formation by regulating *ica* operon manner in *S. aureus*.

### 3.6. ArcR Could Bind to the Promoter Regions of the ica Operon and hla

As a member of the Crp/Fnr family of regulatory proteins, ArcR also contained two typical domains: the N terminus of ArcR contained a circular nucleotide binding domain and the C terminus contained a DNA-binding helix-turn-helix region. To explore whether ArcR could directly bind to the promoter regions of *ica* and *hla*, the binding affinities of ArcR to *ica* (233 bp) and *hla* (283 bp) were analyzed with EMSA. The locations of fragments of *ica* and *hla* are shown in Figure 6A,B. The results of EMSA are shown in Figure 6C,D, the shifted bands emerged and strengthened as the dose of ArcR increased. ArcR did not bind to 267-bp DNA of the *tetA* promoter region (Figure 6E). These results suggested that ArcR could regulate the expression of *icaA* and *hla* by binding to their promoter regions.

## 4. Discussion

Crp/Fnr family regulators are always found in bacteria, and their physiological roles, such as energy metabolism, biofilm formation, antibiotic resistance, and virulence, have been recognized. Previous studies showed that some Crp/Fnr family regulators positively regulated the bacterial virulence and biofilm formation in other bacteria [26,27,28,29,30,31]. In this study, ArcR negatively regulated the biofilm formation and hemolytic activity of *S. aureus*. These differences might be related to the fact that ArcR has low overall similarity with other Crp/Fnr family members, which resulted in a difference in their response to environmental stress. Certainly, ArcR is not an exception. Other Crp/Fnr family members can also repress the expression of functional genes such as Fnr, RedB, DnrD, and DnrF [25,42,43], as the NCTC8325 strain is known to be defective in sigma factor SigB at the stationary phase [44]. Meanwhile, to exclude strain-specific differences, the virulence- and biofilm-associated experiments were repeated with the *arcR* mutant under USA300 background. The deletion of *arcR* in USA300 increased bacterial virulence and biofilm, which was consistent with those in NCTC8325 (Appendix A).

Previous studies have shown that YycFG could directly bind to the promoter regions of *icaA* and positively regulate their expression [23,45]. The promoter region of *yycFG* contains a conserved Crp/Fnr family-like binding motif (-TGTGAttttttaCACA-). To determine whether ArcR regulates the biofilm formation of *S. aureus* via *yycFG*, the expression levels of *yycF* and *yycG* in the WT and Δ*arcR* mutant were evaluated using the results of qRT-PCR and RNA-seq assays. There was no significant difference between the Δ*arcR* mutant and the WT (Appendix A). Taken together, the regulation of ArcR in biofilm formation was not dependent on *yycFG* in our study; however, it is possible that ArcR could regulate the expression of *yycFG* under certain conditions.

The RNA-seq results showed that the virulence-associated genes *sbi*, *vraX,* and *splA* were upregulated (Figure 3B). The second immunoglobulin-binding protein (Sbi) is a multifunctional immune evasion factor that helps *S. aureus* to avoid innate immune defenses [46,47]. VraX is a protein secreted by *S. aureus* and could bind to C1q protein and inhibit the formation of the C1 complex. The deletion of *vraX* reduced the pathogenesis of *S. aureus* [48]. Serine protease-like protein A (SplA) is a member of Spl proteases and is unique to *S. aureus* [49]. SplA could cleave the mucin 16 glycoprotein on the CalU-3 lung cell line, which promotes the spread of *S. aureus* [50]. qRT-PCR and β-galactosidase activity assays further showed that ArcR regulated the expression of *sbi*, *vraX,* and *splA* (Appendix A). These results above suggested that ArcR may globally regulate the virulence of *S. aureus*, rather than affecting hemolysis alone. Therefore, other potential mechanisms by which ArcR regulates virulence and the effect of *arcR* deletion on *S. aureus* virulence in vivo need to be further explored.

It is generally believed that the success of *S. aureus* as a human pathogen is due to its ability to cause a wide range of infections, which requires its various virulence factors and sometimes biofilm formation. Studies on the regulators of virulence and biofilms have shown that the regulatory network of *S. aureus* is highly complex. Thus, further studies are needed to fully understand the molecular mechanisms underlying the regulation of virulence and biofilm formation in *S. aureus*. The regulation of ArcR in *S. aureus* is shown in Figure 7. ArcR binds to the promoter region of *arcABDC* operon and activates its transcription, facilitating the utilization of arginine under anaerobic conditions [32]. ArcR potentially binds to the *yycFG* promoter region and regulates its expression under certain conditions. Our previous study showed that ArcR contributed to tolerance to fluoroquinolone antibiotics by regulating *katA* in *S. aureus*. This study indicated that ArcR could bind to the promoter regions of *ica* operon and *hla*. Here, the physiological roles of ArcR in *S. aureus* NCTC8325 were investigated. These findings may not only provide novel insight into the regulatory networks involved in virulence and biofilm regulation networks, but also expand the physiological roles of ArcR in *S. aureus* and the function of the Crp/Fnr family in pathogenic bacteria.

## Figures and Tables

**Figure 1 microorganisms-11-01656-f001:**
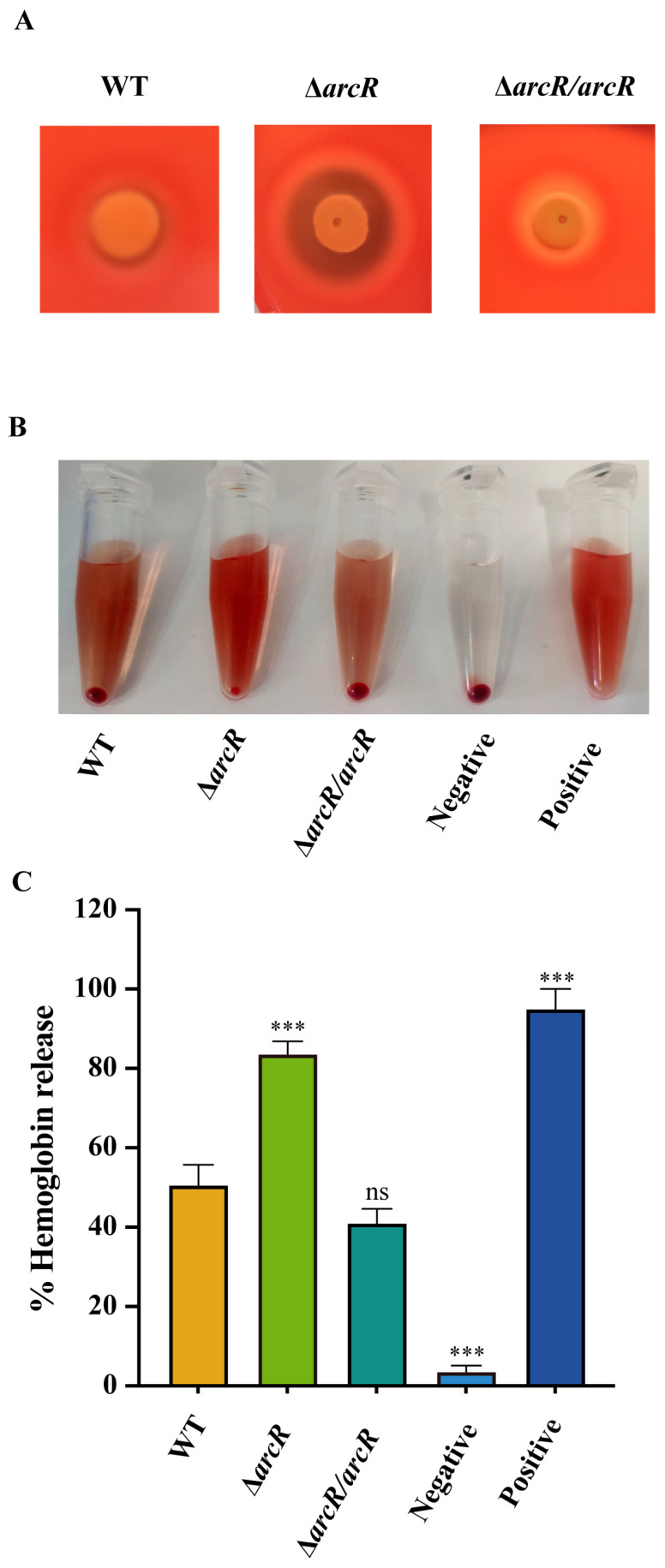
ArcR affects the hemolytic activity of *S. aureus* NCTC8325. (**A**) The hemolytic activities of the WT, Δ*arcR* mutant, and *arcR* complementary strains were performed on SBA plates. (**B**) Hemolytic activities of the WT, Δ*arcR* mutant, and *arcR* complementary strains were determined by incubating samples with 3% sheep red blood cells. PBS, negative control; ddH_2_O, positive control. (**C**) Hemolytic activities of the WT, Δ*arcR* mutant, and *arcR* complementary strains were calculated by measuring the absorption of supernatants at 543 nm. The error bars indicate the standard errors of the means of three biological replicates. *** *p* < 0.001, analyzed with one-way ANOVA.

**Figure 2 microorganisms-11-01656-f002:**
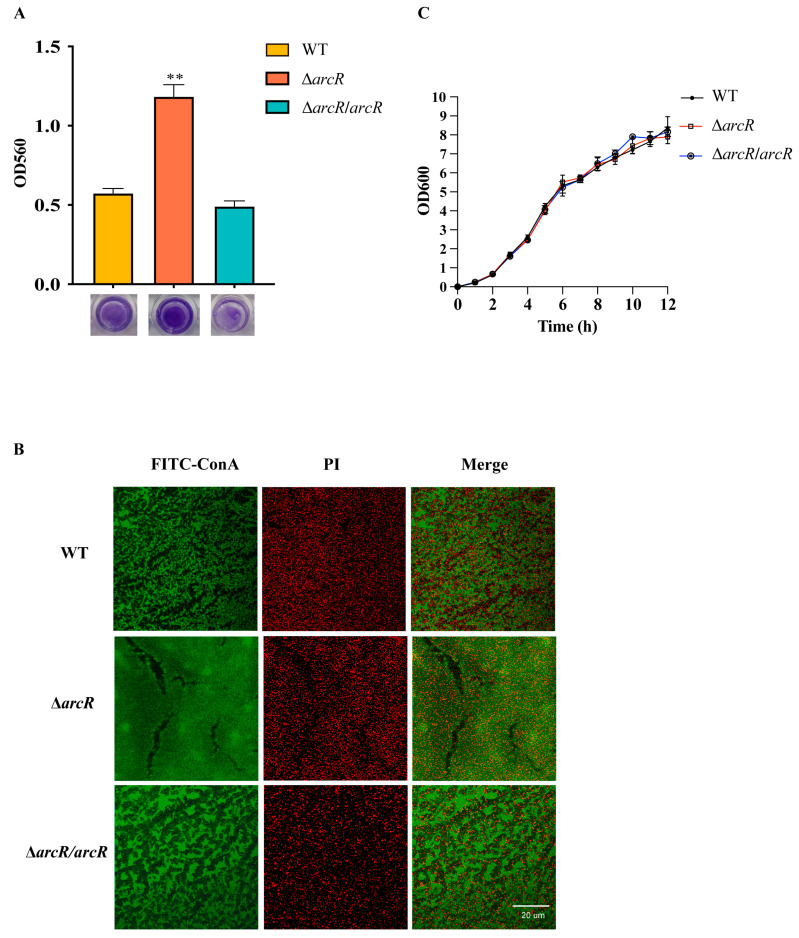
Roles of ArcR in *S. aureus* biofilm formation. The production of biofilm in the WT, Δ*arcR* mutant, and *arcR* complementary strains was detected by 96-well microtiter plate assays (**A**) and CLSM (**B**). CLSM images of the WT, Δ*arcR* mutant, and *arcR* complementary strains’ biofilms were grown on glass-bottom dishes in TSB at 37 °C for 24 h after staining with FITC-ConA (biofilm, green) and PI (bacteria, red). The scale bar represents 20 μm. (**C**) The growth curves of the WT, Δ*arcR* mutant, and *arcR* complementary strains in TSB. The error bars represented the standard errors of the means of three biological replicates. ** *p* < 0.01, analyzed with one-way ANOVA.

**Figure 3 microorganisms-11-01656-f003:**
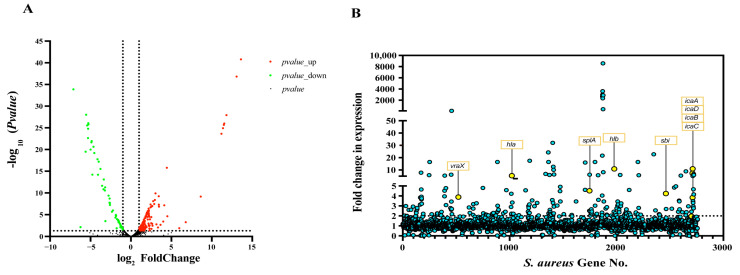
The regulation of ArcR in *S. aureus*. (**A**) Volcano plot depicting gene transcripts to be differentially expressed between the WT and Δ*arcR* mutant. Red and green dots represent significantly upregulated and downregulated proteins, respectively. (**B**) Genes of biofilm and virulence that were remarkably upregulated in the Δ*arcR* mutant.

**Figure 4 microorganisms-11-01656-f004:**
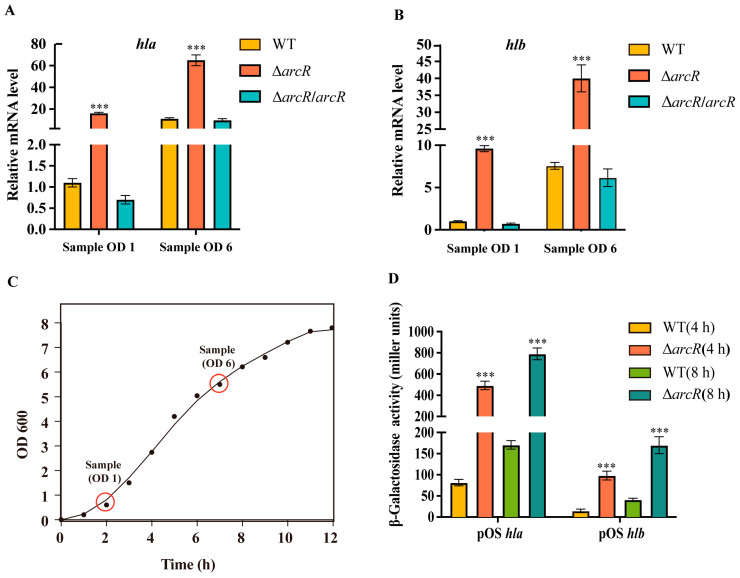
ArcR negatively regulates the transcription of hemolytic genes. The transcript levels of *hla* (**A**), *hlb* (**B**) in the WT, Δ*arcR* mutant, and *arcR* complementary strains at different growth phases. (**C**) The plot with the sample collection points OD1 and OD6. OD1 and OD6 represented OD600 = 1 and OD600 = 6. (**D**) β-Galactosidase activities of *hla* and *hlb* promoter in the WT and Δ*arcR* mutant strains. Bacteria cells were collected at 4 and 8 h. Error bars represent the standard errors of the means of three biological replicates. *** *p* < 0.001, analyzed with one-way ANOVA.

**Figure 5 microorganisms-11-01656-f005:**
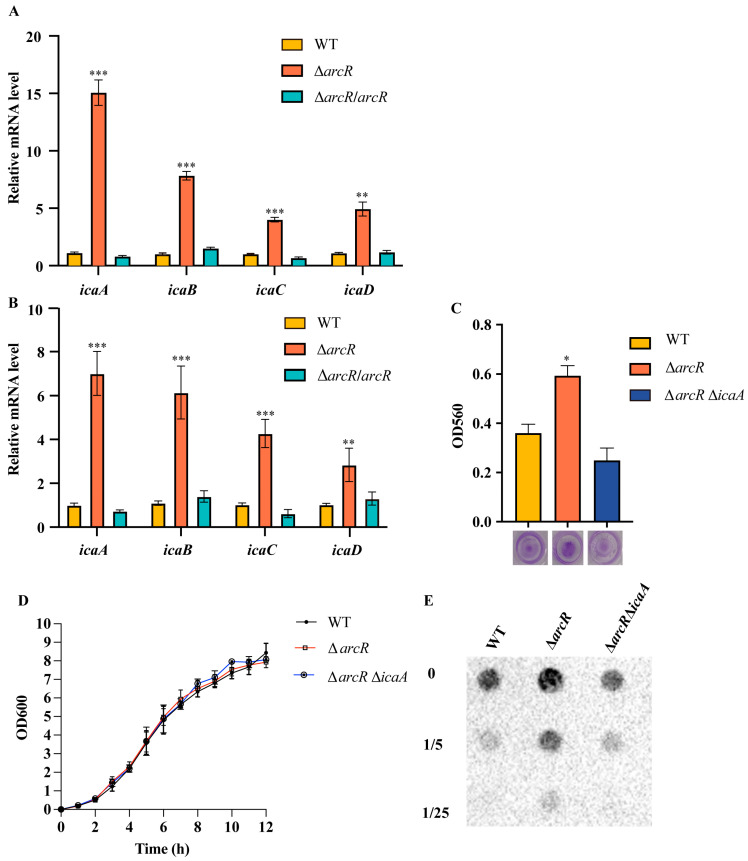
ArcR influences the biofilm formation through *ica* operon. (**A**) The transcript levels of *icaA*, *icaB*, *icaC,* and *icaD* in the WT, Δ*arcR* mutant, and *arcR* complementary strains detected by qRT-PCR in planktonic bacteria (**A**) and in biofilm (**B**). (**C**) Biofilm formation of the WT, Δ*arcR* mutant, and Δ*arcR* Δ*icaA* double mutant strains was detected using 96-well microtiter plate assays. (**D**) A comparison of the growth rates of the WT, Δ*arcR* mutant, and Δ*arcR* Δ*icaA* double mutant strains in TSB. (**E**) The production of PIA in the WT, Δ*arcR* mutant, and Δ*arcR* Δ*icaA*. Samples were diluted at ratios of 1:5 and 1:25. * *p* < 0.05, ** *p* < 0.01, *** *p* < 0.001, analyzed with one-way ANOVA.

**Figure 6 microorganisms-11-01656-f006:**
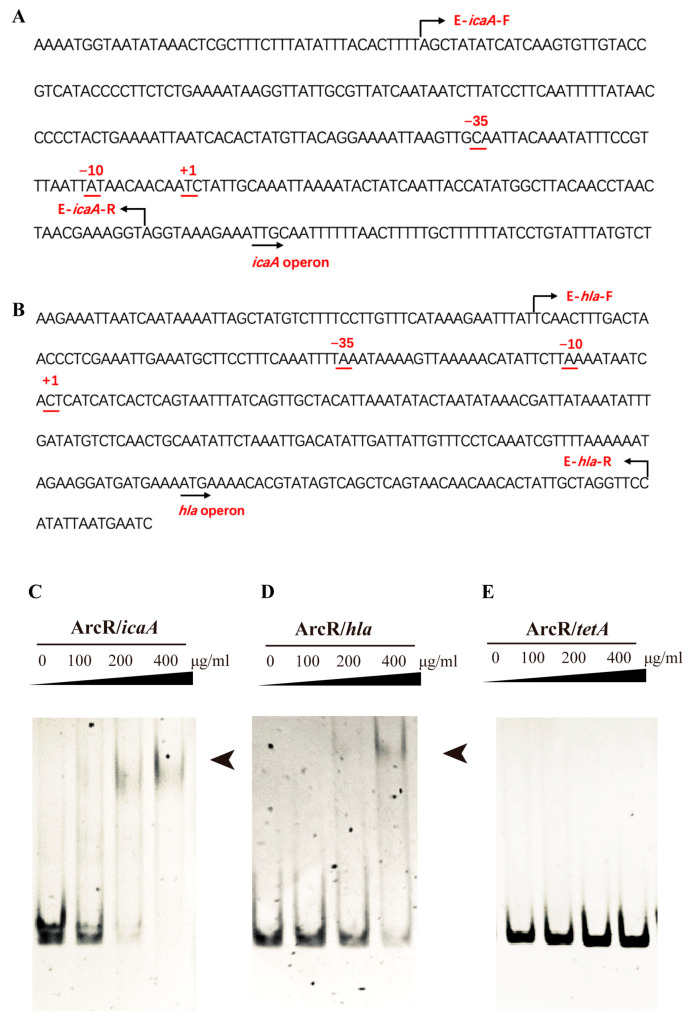
ArcR regulated the expression of the *ica* operon and *hla* by directly binding to their promoter regions. The promoter sequence of the *icaA* and *hla* and the locations of the probes were marked by arrows (**A**,**B**). The interactions between ArcR and the DNA fragments of their promoter regions were detected with EMSA. ArcR was incubated with the *ica* promoter region (**C**), ArcR was incubated with the *hla* promoter region (**D**), and ArcR was incubated with the *tetA* promoter region (**E**) at 37 °C for 30 min. The mixtures were electrophoresed on 8% native polyacrylamide gels and the bands were imaged under UV light after EB staining.

**Figure 7 microorganisms-11-01656-f007:**
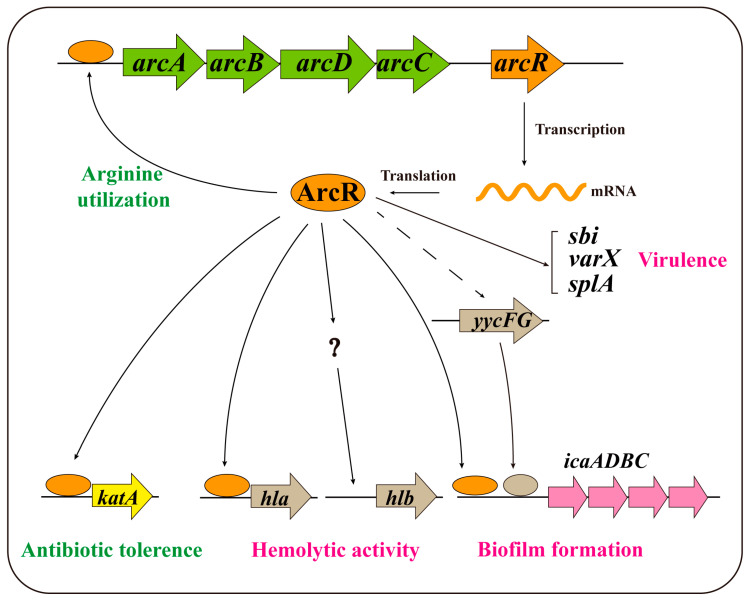
The ArcR regulatory model in *S. aureus*. ArcR binds to the promoter region of the *arcABDC* operon and activates its transcription. ArcR may bind to the *yycFG* promoter region and regulate its expression under certain conditions. Our previous study showed that ArcR contributed to tolerance to fluoroquinolone antibiotics by regulating *katA* in *S. aureus*. Furthermore, ArcR could bind to the promoter regions of *ica* operon and *hla* by directly binding to their promoter regions. ArcR could also participate in the regulation of the virulence-associated genes *sbi*, *vraX,* and *splA*.

**Table 1 microorganisms-11-01656-t001:** Bacterial strains and plasmids used in this study.

Strain or Plasmid	Relevant Characteristic	Source or Reference
*S. aureus*		
RN4220	8325-4, r^−^, restriction-deficient mutagenized RN450	[34]
NCTC8325 WT	NCTC8325 wild-type strain	NASA ^a^
NCTC8325 Δ*arcR*	NCTC8325 *arcR* deletion mutant	[33]
NCTC8325 Δ*arcR* Δ*icaA*	NCTC8325 *arcR icaA* double mutant strain	This study
USA300	CA-MRSA, wild-type	NARSA
USA300 Δ*arcR*	USA300 *arcR* deletion mutant	This study
*E. coli*		
DH5α	*E. coli* host for cloning	Vazyme
BL21 (DE3)	Expression strain; F^−^ *ompT hsdS*_B_ (r_B_^−^ m_B_^−^) *gal dcm* (DE3)	Vazyme
Plasmid		
pBTs	temp-sensitive plasmid, *E. coli* and *S. aureus* shutle vector, Apr, Cmr, for the construction of allelic-exchange mutants	[35]
pBTs-*arcR*	pBTs derivative, for *arcR* deletion	This study
pBTs-*icaA*	pBTs derivative, for *icaA* deletion	This study
pCN51	*E. coli/S. aureus* expression vector, Ap^r^, Em^r^	[36]
pCN51-*arcR*	Complementary *arcR* in *S. aureus*	This study
pET28a	Expression vector with His tag in *E. coli*, Kmr	Addgene
pET28a-*arcR*	His6-ArcR expression vector	This study
pOS1-*lacZ*	*E. coli* and *S. aureus* shuttle vector, with lacZ ORF lacking first 6 amino acids, Ap^r^, Cm^r^.	[37]
pOS1*hla*	pOS1-*lacZ* plasimd derivative, including 546-bp sequence of *hla*	This study
pOS1*hlb*	pOS1-*lacZ* plasimd derivative, including 412-bp sequence of *hlb*	This study
pOS1*sbi*	pOS1-*lacZ* plasimd derivative, including 311-bp sequence of *sbi*	This study
pOS1*vraX*	pOS1-*lacZ* plasimd derivative, including 287-bp sequence of *vraX*	This study
pOS1*splA*	pOS1-*lacZ* plasimd derivative, including sequence of *splA*	This study

^a^, NARSA, network on antimicrobial resistance in *S. aureus*.

## Data Availability

The raw sequence data of RNA-seq in this study have been deposited in the Genome Sequence Archive in National Genomics Data Center, Beijing Institute of Genomics (China National Center for Bioinformation), Chinese Academy of Sciences, under accession number CRA009843, and are publicly accessible at https://bigd.big.ac.cn/gsa, accessed on 16 February 2023.

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
