# Peer review of "Roles of the Crp/Fnr Family Regulator ArcR in the Hemolysis and Biofilm of Staphylococcus aureus"

_microorganisms, 2023, doi:10.3390/microorganisms11071656_

Round 1
Reviewer 1 Report (Previous Reviewer 1)
From the authors’ responses my previous comments:
The RNAseq experiments as presented in the paper display certain genes with a 2-fold change. It does not state that these were the only genes that differed to this extent. Telling me in the response document that the other virulence regulators did not show any alteration without incorporating this information into the manuscript text leaves the reader unsure of this.
The changes to Figure 6 still do not address my question. The pcr fragments are of a different size but migrate identically in the Figure. If the pcr electrophoresis time conditions were different for each, it needs to be indicated. If not, how do you explain the identical migration patterns.
Section 3.2: PI only stains dead cells. This needs to be addressed in this section.
Author Response
- The RNA-seq experiments as presented in the paper display certain genes with a 2-fold change. It does not state that these were the only genes that differed to this extent. Telling me in the response document that the other virulence regulators did not show any alteration without incorporating this information into the manuscript text leaves the reader unsure of this.
Answer:
Thank you for pointing this out. Depends on the RNA-seq, the virulence regulators such as Agr, SigB, SarA, SaeRS and AraC/XylS family protein did not show any significant alteration. We only found that the expression of SarT was upregulated in ΔarcR mutant. But the results of qRT-PCR indicated there was not significant difference on the expression of sarT between WT and ΔarcR mutant.
The upregulated and downregulated genes analyzed by RNA-seq were listed in the Supplementary Table 3 and Table 4.
- The changes to Figure 6 still do not address my question. The pcr fragments are of a different size but migrate identically in the Figure. If the pcr electrophoresis time conditions were different for each, it needs to be indicated. If not, how do you explain the identical migration patterns.
Answer:
We are sorry for the confusion. In EMSA experiment, protein-DNA complexes migrate more slowly than free nucleic acid. The mobility decrement that is observed depends on many factors including the sizes of protein and nucleic acid, the protein charge, the number of protein molecules bound, and so on. In the presence of proteins in different polymerization states such as monomers, dimers, and tetramers in the reaction mixture, multiple binding bands similar to ‘ladder’ may appear (Hellman LM, Fried MG, 2007). In our previous experiments, as well as in the studies of others (Uppal S, Maurya SR, Hire RS, Jawali N, 2011), the size of the DNA fragment has less influence on the blocking distance, but rather the aggregation state of the protein has more influence on the blocking distance. In our experimental results we thought that the aggregation state of ArcR was the same when bound to the promoters of ica and hla, resulting in similar blocking distances. Electrophoresis condition (Limited V, 110V, 2 h, on ice.)
Hellman LM, Fried MG. Electrophoretic mobility shift assay (EMSA) for detecting protein-nucleic acid interactions. Nat Protoc. 2007;2(8):1849-61. doi: 10.1038/nprot.2007.249. PMID: 17703195; PMCID: PMC2757439.
Uppal S, Maurya SR, Hire RS, Jawali N. Cyclic AMP receptor protein regulates cspE, an early cold-inducible gene, in Escherichia coli. J Bacteriol. 2011 Nov;193(22):6142-51. doi: 10.1128/JB.05728-11. Epub 2011 Sep 16. PMID: 21926233; PMCID: PMC3209237.
- Section 3.2: PI only stains dead cells. This needs to be addressed in this section.
Answer:
We are sorry for the confusion. We strongly agreed that PI only stained dead bacteria. In the CLSM experiment, we fixed biofilm with 2.5% glutaraldehyde before stained biofilm bacteria with PI. Fix with 2.5% glutaraldehyde could kill bacteria, change the permeability of cell membranes, and then, PI could stain all the biofilm bacteria cells. In this experiment, PI stained all bacteria in the biofilm. (Shoji Takenaka, et al, 2001)
Takenaka S, Iwaku M, Hoshino E. Artificial Pseudomonas aeruginosa biofilms and confocal laser scanning microscopic analysis. J Infect Chemother. 2001 Jun;7(2):87-93. doi: 10.1007/s101560100014. PMID: 11455498.

Reviewer 2 Report (New Reviewer)
The effect of the regulator ArcR on the hemolysis and biofilm of Staphylococcus aureus was analyzed in the study by Tongtong Fu et al. RNA-seq, gene expression assays, and knockout mutation experiments were carried out. The authors demonstrate that deletion of arcR increased hemolytic ability and biofilm formation as a result of modifications in the regulation of hla, hlb, and genes of ica-operon, respectively. However, no study has been carried out on how ArcR directly impacts virulence. The results of an in vivo mouse infection model (or Galleria mellonella model) and an assessment of the virulence of the NCTC8325 ΔarcR strain would be interesting additions to the study. It should be mentioned in the limitations of the study in the Discussion section.
Minor comments:
«…genes across bacteria.(Kiley and Beinert…» Missing space
«Bacterial rRNA Removal Kit» please specify the vendor.
«High-quality libraries were pooled for Illumina sequencing». What illumina kit was used? What was the number of reads per strain?
The same strain, NCTC8325 ΔarcR, was used by the same authors in the recently published study (10.3389/fmicb.2023.1106340). Do the NCTC8325 ΔarcR strain construction methods used in the current study and those used in previous ones differ in any way? If not, than correct the Table 1 adding the reference in source.
«3.3. Gene expression profiling of the ΔarcR mutant». Please include a Supplementary Table with the list of the upregulated and downregulated genes.
«3.4. ArcR regulates the transcription of hemolytic genes». It would be preferable to include a plot with the sample collection points OD1 and OD6 for easier understanding.
Figure 4. For the abbreviations OD1 and OD6, full names must be given.
Figure 5. The letter (E) is missing in the figure caption.
Author Response
General comments:
The effect of the regulator ArcR on the hemolysis and biofilm of Staphylococcus aureus was analyzed in the study by Tongtong Fu et al. RNA-seq, gene expression assays, and knockout mutation experiments were carried out. The authors demonstrate that deletion of arcR increased hemolytic ability and biofilm formation as a result of modifications in the regulation of hla, hlb, and genes of ica-operon, respectively. However, no study has been carried out on how ArcR directly impacts virulence. The results of an in vivo mouse infection model (or Galleria mellonella model) and an assessment of the virulence of the NCTC8325 ΔarcR strain would be interesting additions to the study. It should be mentioned in the limitations of the study in the Discussion section.
Answer:
We thank the reviewer for the constructive suggestion, which are very important to us. About the question, we offer the point to point answers as below. About the assessment of the virulence of the NCTC8325 ΔarcR strain in vivo, we have added description in the discussion ‘Therefore, other potential mechanisms by which ArcR regulates virulence genes and the effect of arcR deletion on S. aureus virulence in vivo need to be further explored.’ Please refer to lines 376 to 378.
To investigate the effect of arcR deletion on the virulence of S. aureus in vivo, mouse skin infection model was used to compare the virulence of WT, ΔarcR mutant and arcR complementary strains. However, there were not significant and stable results under this model. The effect of arcR deletion on S. aureus virulence in vivo need further investigation. One of the results of mouse skin infection model was shown below.
Minor comments:
1.…genes across bacteria.(Kiley and Beinert) Missing space
Answer:
Thank you for pointing this out, we have added space, please refer to Line 48.
2.Bacterial rRNA Removal Kit, please specify the vendor.
Answer:
Thank you for pointing this out, we have added the vendor ‘Vazyme, Nanjing, China’. Please refer to Lines 148-149.
3.High-quality libraries were pooled for Illumina sequencing. What illumina kit was used? What was the number of reads per strain?
Answer:
Thank you for pointing this out, we have added the description about the illumine kit ‘NEBNext Ultra Directional RNA Library Prep Kit for Illumina (NEB, USA) was used’. Please refer to Lines 157-158.
The number of reads per strain were listed in Table below.
|
Sample name |
Raw reads |
Clean reads |
|
WT |
16640142 |
16496492 |
|
ΔarcR |
15580294 |
15452294 |
4.The same strain, NCTC8325 ΔarcR, was used by the same authors in the recently published study (10.3389/fmicb.2023.1106340). Do the NCTC8325 ΔarcR strain construction methods used in the current study and those used in previous ones differ in any way? If not, than correct the Table 1 adding the reference in source.
Answer:
Thank you for your suggestion, the NCTC8325 ΔarcR strain construction methods used in the current study and those used in previous ones are same. We have added the reference in source in Table 1.
- 3.3. Gene expression profiling of the ΔarcR mutant. Please include a Supplementary Table with the list of the upregulated and downregulated genes.
Answer:
Thank you for pointing this out. The upregulated and downregulated genes were listed in the Supplementary Table 3 and Table 4.
- 3.4. ArcR regulates the transcription of hemolytic genes. It would be preferable to include a plot with the sample collection points OD1 and OD6 for easier understanding.
Answer:
Thank you for your suggestion, we have added a plot with the sample collection points OD1 and OD6 in Figure 4. Please refer to Figure 4C.
- Figure 4. For the abbreviations OD1 and OD6, full names must be given.
Answer:
Thank you for pointing this out. The full name ‘optical density’ was added, when it appeared for the first time. Please refer to line 124. We also added description about OD1 and OD6 at Figure 4 legend as ‘OD1 and OD6 represented OD600=1 and OD600=6’. Please refer to lines 293-294.
- Figure 5. The letter (E) is missing in the figure caption.
Answer:
We are sorry for the confusion. It is possible that the figure caption ‘E’ is too far away from the picture causing such a misunderstanding. We repositioned ‘E’ so that it is at a more appropriate distance from the picture. At the same time, we moved the place of the annotation of ‘E’ in Figure 5 legend. Please refer to line 321.

Reviewer 3 Report (New Reviewer)
This is a significant study and I appreciate the opportunity to review it. I have few suggestions since it is a coherent text and that makes perfect sense.
For my part, I suggest that in item 3.1 where the quantitative hemolytic activity is addressed, that the text contain the numbers and not just the graphs.
I also think it's important to stress the importance of the B-galactosidase analyzes in the Introduction. It was surprising to see the method without its importance being related.
Author Response
This is a significant study and I appreciate the opportunity to review it. I have few suggestions since it is a coherent text and that makes perfect sense.
For my part, I suggest that in item 3.1 where the quantitative hemolytic activity is addressed, that the text contain the numbers and not just the graphs.
I also think it's important to stress the importance of the B-galactosidase analyzes in the Introduction. It was surprising to see the method without its importance being related.
Answer:
We thank the reviewer for the constructive suggestion, which are very important to us. About the question:
1) We have added the numbers about the quantitative hemolytic activity in the Supplementary Table 2.
2) We have added the description of the importance of the β-galactosidase analyzes in the Introduction as below ‘RNA-Seq and qRT-PCR assays indicated that icaADBC, hla and hlb was upregulated in the ΔarcR mutant compared with the wild type (WT). The β-Galactosidase activity assay provides more direct evidence of the role of ArcR in the regulation of genes in S. aureus. The results of β-Galactosidase activity assay showed that the deletion of arcR increased the promoter activity of hla and hlb. The construction of ΔarcR ΔicaA double mutant strain and the detection of PIA production indicated that arcR increased the biofilm formation through ica operon. Further study demonstrated that ArcR could bind to the promoter regions of hla and ica operon promoter regions and regulate their expression.’ Please refer to lines 62-74.

This manuscript is a resubmission of an earlier submission. The following is a list of the peer review reports and author responses from that submission.
Round 1
Reviewer 1 Report
The authors created mutants lacking the transcriptional regulator ArcR and demonstrated that this mutant exhibits elevated hemolytic activity and biofilm formation. RNAseq results show an increased expression of the ica genes (biofilm associated) and hla/hlb (alpha and beta hemolysins). EMSA experiments showed band shifting of hla and ica promoter region fragments. ArcR is a known S. aureus transcriptional regulator that is upregulated in biofilms. This seems at odds with the authors’ findings that ArcR is a negative regulator of biofilm formation. It is possible that ArcR’s effects on these two phenotypes is a result of its regulation of the arginase deaminase operon and effects on biofilm and hemolysis are indirect. The EMSA results would suggest otherwise, but as pointed out below, there is a potential flaw in the EMSA methodology. The direct role of ArcR in regulation of these genes cannot be assumed, since it is well known that the transcriptional regulators of S. aureus often impact expression of other regulators in the network. It is also important to amend the manuscript to emphasize that the experiments were not performed on biofilm cells, but on planktonic cultures. The statement that Propidium iodide stains the bacterial cells needs to be changed. It is a membrane impermeant dye, so it only stains dead cells. Thus what is shown in Figure 2 are dead cells, not total cells. The marked increase in dead cells in the arcR mutant culture may contribute to changes in amount of biofilm formed. Also, a marked increase in dead cells could impact the interpretation of the results of the RNAseq and qRT PCR experiments.
Specific comments:
1. Introduction line 9: What is PFT?
2. Introduction line 14: These “do regulate”, not “could regulate”.
3. Page 2, line 9: Since S. aureus can cause bacteremia, they are not ALWAYS attached to medical devices or host tissues.
4. Page 2, line 11: How does biofilm formation increase host defense?
5. Page 2, last paragraph: This is not actual introduction material, but a repeat of the abstract. What is lacking in the Introduction section is a discussion of what is known about ArcR in Staphylococcus aureus (namely arginase deaminase operon regulation). Additionally, there are two publications (the second listed below was cited in the manuscript) that show that the arcR gene is upregulated in biofilms. How do you explain your findings that ArcR is a negative regulator of biofilm ica genes in this context?
Resch, A., R. Rosenstein, C. Nerz, and F. Gotz. 2005. Differential gene expression profiling of Staphylococcus aureus cultivated under biofilm and planktonic conditions. Appl. Environ. Microbiol. 71:2663-2676. https://www.ncbi.nlm.nih.gov/pmc/articles/PMC1087559/
Beenken, K. E., P. M. Dunman, F. McAleese, D. Macapagal, E. Murphy, S. J. Projan, J. S. Blevins, and M. S. Smeltzer. 2004. Global gene expression in Staphylococcus aureus biofilms. J. Bacteriol. 186:4665-4684. https://www.ncbi.nlm.nih.gov/pmc/articles/PMC438561/
6. Figure 1: There seems to be inconsistencies in the results. In panel A, the hemolytic activity of the complemented strain appears to be much greater than that of the wild-type parent strain, yet in panel C, the complemented activity is less than that of the parent strain. This raises the question as to how consistent the hemolytic or biofilm phenotypes are (only one colony is shown in Fig. 1, for example).
7. Page 7 top paragraph and Figure 2B: The PI image is so faint, that the merged images are essentially the same as the FITC image, and thus do not contribute any useful information. It is also misleading to state that PI stains bacteria. It only stains the dead cells. You observed an increase in dead cells with the mutant strain, which does not seem to be complemented with the plasmid copy of the gene (although better quantification of the images would be helpful). Because it is not possible to assess biofilm based on the FITC staining pattern between the wildtype and the mutant, it is not possible to reconcile the reduction in biofilm based on crystal violet staining (panel 2A) and the similar conA staining patters shown in panel 2B. This makes it difficult to assess the actual contribution of ArcR to biofilm formation. Perhaps, the reduction in biofilm is simply due to the reduced viability of the cells.
8. Section 3.3: The methodology used in the RNAseq experiment examined gene expression patterns in planktonic cells, but not in biofilm conditions. So under these conditions, it appears that ArcR downregulates the ica genes. Again, results from other laboratories have shown that the arcR gene is upregulated in biofilms which seems counterintuitive to the results presented here. This should be addressed in the manuscript.
9. Figure 4D makes no sense to me. There is no indication in the literature that hemolysins are induced in the presence of erythrocytes, so the results are obvious. Also, it is not indicated in the figure legend or in the methods at what growth stage the cells were exposed to the RBCs, so one cannot compare the results from 4B to those of 4A and B. The RBC exposure methodology is missing in the Methods section.
10. Figure 5 legend: There is no panel E.
11. Section 3-5: The ica gene transcription is downregulated by ArcR and biofilm is increased. Thus there is a correlation between the two. However, it is not possible to conclude that the biofilm effect is dependent on the ica gene levels. Biofilm formation is dependent on many genes, not just the ica genes. To show dependency, you would have to show that the ica gene expression alone can reverse the biofilm phenotype. Again, it should be pointed out that the PIA was extracted from planktonic cells, not from biofilm cells. Also, these results only show correlation, not proof of dependency.
12. Section 3-6: I have two concerns with the EMSA experiments. One, I do not know what the tetA gene is that was used as the negative control. Is it an S. aureus gene? There is no mention of it in the methods section and no mention in the data bases that I searched. Second, S. aureus transcriptional factors are well known to have nonspecific DNA binding activity in addition to the specific target sequences. This can be ruled out by including an excess of sonicated nonspecific DNA, such as salmon sperm DNA, to the EMSA buffer.
13. Section 3-6: The matches of the sequences you identified in the ica, hla, and hlb promoter regions are very poor compared with the consensus sequence, both in terms of nucleotide matches and spacing between the two arms of the element. The hla spacing is only 2 bp, which markedly changes the exposure of the sequence to the ArcR protein. It appears that you used capital letters to denote matches to the consensus sequence, but when the sequences are aligned, this is not the case. What is the basis of the upper versus lower case lettering?
(-TGTGA-N6-TCACA-)
(-TCAtAccccttcTcTGA-) ica
(-TGAtAtgTcTCA-) hla
(-TGTcgAttataTttACA-) hlb
14. Discussion: Not all of the roles of ArcR are unknown, such as its role in arginase deaminase expression. Also, the direct binding experiments are potentially flawed (as noted above) and any direct effects of ArcR on transcription are speculative. You have not shown that ArcR binds to the target sequence or that it can inhibit transcription in vitro. Your data may suggest this, but proof is still lacking. Your results do strongly indicate that there is a correlation between loss of ArcR and increased transcription of the ica operon and the hla and hlb genes, but whether it is a direct effect or an indirect effect is unknown.
Author Response
A point-by-point response to Reviewer #1 Comments:
Comments and Suggestions for Authors
The authors created mutants lacking the transcriptional regulator ArcR and demonstrated that this mutant exhibits elevated hemolytic activity and biofilm formation. RNAseq results show an increased expression of the ica genes (biofilm associated) and hla/hlb (alpha and beta hemolysins). EMSA experiments showed band shifting of hla and ica promoter region fragments. ArcR is a known S. aureus transcriptional regulator that is upregulated in biofilms. This seems at odds with the authors’ findings that ArcR is a negative regulator of biofilm formation. It is possible that ArcR’s effects on these two phenotypes is a result of its regulation of the arginase deaminase operon and effects on biofilm and hemolysis are indirect. The EMSA results would suggest otherwise, but as pointed out below, there is a potential flaw in the EMSA methodology. The direct role of ArcR in regulation of these genes cannot be assumed, since it is well known that the transcriptional regulators of S. aureus often impact expression of other regulators in the network. It is also important to amend the manuscript to emphasize that the experiments were not performed on biofilm cells, but on planktonic cultures. The statement that Propidium iodide stains the bacterial cells needs to be changed. It is a membrane impermeant dye, so it only stains dead cells. Thus what is shown in Figure 2 are dead cells, not total cells. The marked increase in dead cells in the arcR mutant culture may contribute to changes in amount of biofilm formed. Also, a marked increase in dead cells could impact the interpretation of the results of the RNAseq and qRT PCR experiments.
Answer:
We thank the reviewer for the constructive suggestion. About the question, we offer the point to point answers as below. We added description in discussion ‘The RNA-seq and qRT-PCR results showed that ArcR regulated the expression of ica operon and subsequently influenced the production of PIA in planktonic cells. PIA is an important adherent agent and helps planktonic cells adhesion and then influences the biofilm formation. These experiments were performed in planktonic cells, the regulation of ArcR to ica operon in biofilm needs further studying.’ to emphasize that the experiments were not performed on biofilm cells, but on planktonic cultures.
Specific comments:
- Introduction line 9: What is PFT?
Answer:
Thanks for pointing this out, the whole word of ‘PFT’ is ‘pore-forming toxins’. We have changed the wording. Please refer to line 50
- Introduction line 14: These “do regulate”, not “could regulate”.
Answer:
Thanks for pointing this out, correction has been made as suggested. Please refer to line 54.
- Page 2, line 9: Since S. aureus can cause bacteremia, they are not ALWAYS attached to medical devices or host tissues.
Answer:
Thanks for pointing this out, we have removed ‘always’. Please refer to lines 62-63.
- Page 2, line 11: How does biofilm formation increase host defense?
Answer:
Biofilm plays an important role during infection by providing a defense against several clearance mechanisms. The biofilm matrix can impede the access of certain types of immune defenses, such as macrophages, which display incomplete penetration into the biofilm matrix and “frustrated phagocytosis” (Scherr et al., 2014). Beyond these mechanisms, biofilms also play an important role in the progression of chronic diseases. Following the establishment of a biofilm, individual cells can disperse from the original biofilm and either seed new sights of infection or mediate an acute infection such as sepsis (Costerton et al., 1999).
- Page 2, last paragraph: This is not actual introduction material, but a repeat of the abstract. What is lacking in the Introduction section is a discussion of what is known about ArcR in Staphylococcus aureus (namely arginase deaminase operon regulation). Additionally, there are two publications (the second listed below was cited in the manuscript) that show that the arcR gene is upregulated in biofilms. How do you explain your findings that ArcR is a negative regulator of biofilm ica genes in this context?
Resch, A., R. Rosenstein, C. Nerz, and F. Gotz. 2005. Differential gene expression profiling of Staphylococcus aureus cultivated under biofilm and planktonic conditions. Appl. Environ. Microbiol. 71:2663-2676. https://www.ncbi.nlm.nih.gov/pmc/articles/PMC1087559/
Beenken, K. E., P. M. Dunman, F. McAleese, D. Macapagal, E. Murphy, S. J. Projan, J. S. Blevins, and M. S. Smeltzer. 2004. Global gene expression in Staphylococcus aureus biofilms. J. Bacteriol. 186:4665-4684. https://www.ncbi.nlm.nih.gov/pmc/articles/PMC438561/
Answer:
Thanks for pointing this out, we have rewritten the last paragraph of introduction. Please refer to lines 347-352
The expression of arcR did not change in biofilm in the two articles you mentioned. It was reported that arcR and arcA belong to the same operon, but it is found in Northern blot experiments that there may be different transcripts of arcR, so the expression of arcR may not always be consistent with that of arcA. (Julia Makhlin, Tzili Kofman, et al. 2007). So, the expression of arcR in biofilm still need further studying. The transcriptome results of these two articles are not consistent and cannot be verified by each other. Transcriptome results can be influenced by many conditions and ultimately need to be verified by experiments. We added description in discussion ‘The RNA-seq and qRT-PCR results showed that ArcR regulated the expression of ica operon and subsequently influenced the production of PIA in planktonic cells. PIA is an important adherent agent and helps planktonic cells adhesion and then influences the biofilm formation. These experiments were performed in planktonic cells, the regulation of ArcR to ica operon in biofilm needs further studying.’
- Figure 1: There seems to be inconsistencies in the results. In panel A, the hemolytic activity of the complemented strain appears to be much greater than that of the wild-type parent strain, yet in panel C, the complemented activity is less than that of the parent strain. This raises the question as to how consistent the hemolytic or biofilm phenotypes are (only one colony is shown in Fig. 1, for example).
Answer:
Thanks for pointing this out, there was no significant difference in hemolysis activity between wild-type strain and supplement strain in Fig.1 C. We did three biological replicates for every experiments. The hemolytic and biofilm phenotypes are consistent. In order to reduce misunderstanding, we changed the picture of the complementary strain in Figure 1A
- Page 7 top paragraph and Figure 2B: The PI image is so faint, that the merged images are essentially the same as the FITC image, and thus do not contribute any useful information. It is also misleading to state that PI stains bacteria. It only stains the dead cells. You observed an increase in dead cells with the mutant strain, which does not seem to be complemented with the plasmid copy of the gene (although better quantification of the images would be helpful). Because it is not possible to assess biofilm based on the FITC staining pattern between the wildtype and the mutant, it is not possible to reconcile the reduction in biofilm based on crystal violet staining (panel 2A) and the similar conA staining patters shown in panel 2B. This makes it difficult to assess the actual contribution of ArcR to biofilm formation. Perhaps, the reduction in biofilm is simply due to the reduced viability of the cells.
Answer:
Thanks for pointing this out, we have remerged the images as suggested. In the CLSM experiment, we fixed biofilm with 2.5% glutaraldehyde before stained biofilm bacteria with PI. Fix with 2.5% glutaraldehyde could kill bacteria, change the permeability of cell membranes, and then, PI could stain the cells. In this experiment, PI stained almost all bacteria in the biofilm. (Shoji Takenaka, et al, 2001)
- Section 3.3: The methodology used in the RNAseq experiment examined gene expression patterns in planktonic cells, but not in biofilm conditions. So under these conditions, it appears that ArcR downregulates the ica genes. Again, results from other laboratories have shown that the arcR gene is upregulated in biofilms which seems counterintuitive to the results presented here. This should be addressed in the manuscript.
Answer:
Thanks for pointing this out, we added description in discussion ‘The RNA-seq and qRT-PCR results showed that ArcR regulated the expression of ica operon and subsequently influenced the production of PIA in planktonic cells. PIA is an important adherent agent and helps planktonic cells adhesion and then influences the biofilm formation. These experiments were performed in planktonic cells, the regulation of ArcR to ica operon in biofilm needs further studying.’
- Figure 4D makes no sense to me. There is no indication in the literature that hemolysins are induced in the presence of erythrocytes, so the results are obvious. Also, it is not indicated in the figure legend or in the methods at what growth stage the cells were exposed to the RBCs, so one cannot compare the results from 4B to those of 4A and B. The RBC exposure methodology is missing in the Methods section.
Answer:
Thanks for pointing this out, we wanted to show that the expression differences of hla and hlb between WT and arcR mutant also existed under RBC exposure condition. Sorry for the confusion, we have changed the wording. Please refer to lines 302-303.
- Figure 5 legend: There is no panel E.
Answer:
Thanks for pointing this out, we removed panel E as suggested. Please refer to line 573.
- Section 3-5: The ica gene transcription is downregulated by ArcR and biofilm is increased. Thus there is a correlation between the two. However, it is not possible to conclude that the biofilm effect is dependent on the ica gene levels. Biofilm formation is dependent on many genes, not just the ica genes. To show dependency, you would have to show that the ica gene expression alone can reverse the biofilm phenotype. Again, it should be pointed out that the PIA was extracted from planktonic cells, not from biofilm cells. Also, these results only show correlation, not proof of dependency.
Answer:
Thanks for pointing this out, our RNA-seq results showed that the expression level of ica operon was up-regulated in arcR mutant. In ΔarcR mutant, the ica deletion will reduce the amount of biofilm, which further indicates that arcR regulated the biofilm formation by an ica dependent manner. We added description in discussion ‘The RNA-seq and qRT-PCR results showed that ArcR regulated the expression of ica operon and subsequently influenced the production of PIA in planktonic cells. PIA is an important adherent agent and helps planktonic cells adhesion and then influences the biofilm formation. These experiments were performed in planktonic cells, the regulation of ArcR to ica operon in biofilm needs further studying.’
- Section 3-6: I have two concerns with the EMSA experiments. One, I do not know what the tetA gene is that was used as the negative control. Is it an S. aureus gene? There is no mention of it in the methods section and no mention in the data bases that I searched. Second, S. aureus transcriptional factors are well known to have nonspecific DNA binding activity in addition to the specific target sequences. This can be ruled out by including an excess of sonicated nonspecific DNA, such as salmon sperm DNA, to the EMSA buffer.
Answer:
Thanks for pointing this out, the EMSA experiment does have some limitations, but we set the concentration gradient, negative fragment control and negative protein control in the experiment, which is enough to show that the binding of protein and fragment we measured is specific. tetA fragment and protein TetR21 could bound with each other reported in our previous studies, and were used as controls in the EMSA experiment. tetA gene number is MW2296.
- Section 3-6: The matches of the sequences you identified in the ica, hla, and hlb promoter regions are very poor compared with the consensus sequence, both in terms of nucleotide matches and spacing between the two arms of the element. The hla spacing is only 2 bp, which markedly changes the exposure of the sequence to the ArcR protein. It appears that you used capital letters to denote matches to the consensus sequence, but when the sequences are aligned, this is not the case. What is the basis of the upper versus lower case lettering?
(-TGTGA-N6-TCACA-)
(-TCAtAccccttcTcTGA-) ica
(-TGAtAtgTcTCA-) hla
(-TGTcgAttataTttACA-) hlb
Answer:
Thanks for pointing this out, capital letters indicate that the base is consistent with the base of predicted binding motif at current position, and lower cases indicate that it is inconsistent. We speculate that these sequences may be the binding sequences of ArcR.
- Discussion: Not all of the roles of ArcR are unknown, such as its role in arginase deaminase expression. Also, the direct binding experiments are potentially flawed (as noted above) and any direct effects of ArcR on transcription are speculative. You have not shown that ArcR binds to the target sequence or that it can inhibit transcription in vitro. Your data may suggest this, but proof is still lacking. Your results do strongly indicate that there is a correlation between loss of ArcR and increased transcription of the ica operon and the hla and hlb genes, but whether it is a direct effect or an indirect effect is unknown.
Answer:
Thanks for pointing this out, in the discussion part, we changed the description of ArcR function as suggested. The EMSA experiment does have some limitations, but we set the concentration gradient, negative fragment control and negative protein control in the experiment, which is enough to show that the binding of protein and fragment we measured is specific. The EMSA experiments indicated that ArcR could bind to the promoter regions of ica, hla and hlb. It was proved that ArcR directly regulated the expression of these genes.

Reviewer 2 Report
Dear authors,
the manuscript presents interesting results for readers. The experiments are well-designed and the results support the conclusions.
However, I suggest deep text editing.
Please, rephrase the sentences:
-S. aureus always attached to medical devices and host tissues and forming a mature biofilm.
-Crp/Fnr family regulators also plays
-Lmo0753 is involved in the biofilm formation and attachment to fresh
produce
in the introduction, the sentences "The mutation of arcR increased the hemolytic ability by the upregulation of hla and hlb. Meanwhile, the deletion of arcR promoted the biofilm formation. RNA-Seq and real-time quantitative reverse transcription PCR (qRT-PCR) assays indicated that the expression of icaADBC was upregulated in the .arcR mutant compared with the wild type (WT)." anticipate the results.
-Antibiotics were used for plasmids selection
-The ΔarcR ΔicaA double mutant strains was constructed
-The shuttle vector pCN51 was used to constructed complementary strains.
-Then observe the film around the wall and bottom of the wells to confirm the biofilm production.
- why an enzyme-linked immunosorbent assay reader was used?
-the sentence "Confocal laser scanning microscope (CLSM) has been introduced for use in analysis to provide a better understanding of biofilm structure." is too naive.
-avoid repetition "and then the bacterial organisms in the biofilms were then visualized", "extracted PIA extracted", "we examined the biofilm architecture and exopolysaccharides quantity on glass bottomed cell culture dish were examined."
-use 1 ml instead of 1,000 µl along with the text
-pay attention to the words ribosome RNA, the first cDNA, PH, To explore whether ArcR could directly bound to the promoter region, ArcR binds to the promoter region of arcABDC operon and activates its transcription (if a proteins binds to the promoter, it should work as a repressor)
-enzymes EcoR1 and BamH1 in italics
-how many proteins were prepared? I read "These fusion proteins"
-please use the whole words the first time you use them, i.e PFT, PI, TSB, PBS and so on.
Author Response
A point-by-point response to Reviewer #2 Comments:
Comments and Suggestions for Authors
Dear authors,
the manuscript presents interesting results for readers. The experiments are well-designed and the results support the conclusions.
Answer:
We thank the reviewer for the constructive suggestion.
However, I suggest deep text editing.
Answer:
Thanks for pointing this out, as suggested, the manuscript has been edited by an English native speaker to ensure language and grammar accuracy. We have sent an editing certificate to the Editorial office.
Please, rephrase the sentences:
-S. aureus always attached to medical devices and host tissues and forming a mature biofilm.
Answer:
As suggested, the sentence has been rephrased to ‘S. aureus attaches to medical devices and host tissues which contributes to the formation of biofilm.’ Please refer to lines 62-63.
-Crp/Fnr family regulators also plays
Answer:
As suggested, correction has been made. ‘Plays’ has been replaced by ‘play’ Please refer to line 86.
-Lmo0753 is involved in the biofilm formation and attachment to fresh produce
Answer:
Thanks for pointing this out, ‘produce’ has been replaced by ‘agricultural products’. Please refer to line 91.
in the introduction, the sentences "The mutation of arcR increased the hemolytic ability by the upregulation of hla and hlb. Meanwhile, the deletion of arcR promoted the biofilm formation. RNA-Seq and real-time quantitative reverse transcription PCR (qRT-PCR) assays indicated that the expression of icaADBC was upregulated in the .arcR mutant compared with the wild type (WT)." anticipate the results.
Answer:
Thanks for pointing this out, we have rewritten this paragraph. Please refer to lines 93-98.
-Antibiotics were used for plasmids selection
Answer:
Sorry for the confusion, we add ‘constructed’ before ‘plasmids’. Please refer to line 106.
-The ΔarcR ΔicaA double mutant strains was constructed
Answer:
Thanks for pointing this out, correction has been made. We have replaced ‘strains’ with ‘strain’ Please refer to line 111.
-The shuttle vector pCN51 was used to constructed complementary strains.
Answer:
Thanks for pointing this out, correction has been made. We have replaced ’constructed’ with ‘construct’. Please refer to line 118.
-Then observe the film around the wall and bottom of the wells to confirm the biofilm production.
Answer:
Thanks for pointing this out, we have rewritten this sentence as below ‘Then, the film around the wall and bottom of the wells was observed to confirm biofilm production.’ Please refer to lines 127-128.
- why an enzyme-linked immunosorbent assay reader was used?
Answer:
It was used to detect the absorbance at 540 nm.
-the sentence "Confocal laser scanning microscope (CLSM) has been introduced for use in analysis to provide a better understanding of biofilm structure." is too naive.
Answer:
Thanks for pointing this out, we have rewritten this sentence as below ‘Confocal laser scanning microscopy (CLSM) has been introduced for observing the biofilm production and form.’ Please refer to lines 135-136.
-avoid repetition "and then the bacterial organisms in the biofilms were then visualized", "extracted PIA extracted", "we examined the biofilm architecture and exopolysaccharides quantity on glass bottomed cell culture dish formexamined."
Answer:
As suggested, we have removed the repeat words.
-use 1 ml instead of 1,000 µl along with the text
Answer:
As suggested, correction has been made.
-pay attention to the words ribosome RNA, the first cDNA, PH, To explore whether ArcR could directly bound to the promoter region, ArcR binds to the promoter region of arcABDC operon and activates its transcription (if a proteins binds to the promoter, it should work as a repressor)
Answer:
As suggested, ‘ribosome RNA’ replaced by ‘rRNA’. ‘PH’ replaced by ‘pH’. ‘the first cDNA’ replaced by ‘First strand of cDNA’.
ArcR binds to the promoter region of arcABDC operon and activates its transcription.( Julia Makhlin, Tzili Kofman, 2007) A proteins binds to the promoter, it could also work as a activator.
-enzymes EcoR1 and BamH1 in italics
Answer:
As suggested, correction has been made. Please refer to line 193.
-how many proteins were prepared? I read "These fusion proteins"
Answer:
As suggested, ‘These fusion proteins’ replaced by ‘This fusion protein’ Please refer to line 234.
-please use the whole words the first time you use them, i.e PFT, PI, TSB, PBS and so on.
Answer:
As suggested, correction has been made.

Reviewer 3 Report
Figure 1C - you represent the % of haemolysis not the % haemoglobin release
What the practical relevance of your study?
Author Response
A point-by-point response to Reviewer #3 Comments:
Comments and Suggestions for Authors
Figure 1C - you represent the % of haemolysis not the % haemoglobin release
What the practical relevance of your study?
Answer:
Thanks for pointing this out, hemolysins are important virulence factors of Staphylococcus aureus, which can lysate red blood cells. It is much more straightforward and simpler to measure the number of ruptured red blood cells than to measure the amount of hemoglobin released.

Round 2
Reviewer 1 Report
The authors have shown that loss of ArcR results in increased expression of the two hemolysin genes and the ica polysaccharide adhesion genes. The EMSA experiments demonstrate that the ArcR protein can bind to DNA fragments from the promoter regions of the hla, hlb, and icaA genes. From this, they claim that ” It was proved that ArcR directly regulated the expression of these genes”. The problem remains that they have not definitively shown that 1) the ArcR protein bind specifically to the target sequences indicated (which are only a small part of the icaA 236 bp fragment, for example). Furthermore, there is no evidence (for or against) this binding directly affecting transcription. Direct negative regulation by ArcR on these genes is one possible interpretation. Others would be a indirect effect resulting from down-regulation of regulatory networks that up-regulate the hemolysins or ica genes or up-regulation of networks that down-regulate them.
Section 3.3: The RNAseq results do not address effects on the Agr, or SarA, SarX, and Rfb networks that impact the hemolysin and/or ica operon expression. What was the impact of the arcR deletion on expression of these regulators?
Section 3.4: The roles of Hla and Hlb in hemolysis are well defined both in terms of mode of action and regulation of their genes. The experiment involving incubation of the bacteria in the presence of blood is not defined as to what the outcome options would be and so how this would further define their roles.
Section 3.5: There is no data from an arcR icaA double deletion mutant shown in Figure 5, except for the growth curve in Figure 5C. Thus it is not possible to evaluate the claim of PIA-dependence.
Section 3.6 and Figure 6: The target DNA fragment sizes are listed as ica (233 bp), hla (283 bp) and hlb (251 bp), the mobility of these appear to be inconsistent with this. Were these gels electrophoresed for different times? If so, this should be indicated in the figure legend.
Author Response
A point-by-point response to Reviewer Comments:
The authors have shown that loss of ArcR results in increased expression of the two hemolysin genes and the ica polysaccharide adhesion genes. The EMSA experiments demonstrate that the ArcR protein can bind to DNA fragments from the promoter regions of the hla, hlb, and icaA genes. From this, they claim that ” It was proved that ArcR directly regulated the expression of these genes”. The problem remains that they have not definitively shown that 1) the ArcR protein bind specifically to the target sequences indicated (which are only a small part of the icaA 236 bp fragment, for example). Furthermore, there is no evidence (for or against) this binding directly affecting transcription. Direct negative regulation by ArcR on these genes is one possible interpretation. Others would be a indirect effect resulting from down-regulation of regulatory networks that up-regulate the hemolysins or ica genes or up-regulation of networks that down-regulate them.
Answer:
We thank the reviewer for the constructive suggestion. In the 3.6 part of the results of manuscript, we deleted the prediction of possible binding sequences of hla, hlb, and icaA. We fully agree with you that the regulatory network affect the expression of these genes, and ArcR may also regulate the expression of them through other indirect ways. However, there was no significant change about the expression of Agr, SarA, SarX, and Rbf in our transcriptome results (Fold change log2 > 2 is considered significant change). We used β-galactosidase activity experiments to detect the promoter activities of hla and hlb. The β-galactosidase activities in the ΔarcR mutant strain were significantly higher than those in the WT strain. These results could indicate these binding between ArcR and these genes affected their transcription to a certain extent. But it's true that there's no particularly direct evidence that these bindings affect the transcription of these genes. We have changed the wording from ‘directly regulate these genes expression’ to ‘could bind to their promoter regions’ and added discussion about EMSA experiments. Meanwhile, further experiments are needed to prove whether ArcR can regulate the expression of these genes in other indirect ways.
Section 3.3: The RNAseq results do not address effects on the Agr, or SarA, SarX, and Rbf networks that impact the hemolysin and/or ica operon expression. What was the impact of the arcR deletion on expression of these regulators?
Answer:
Thanks for pointing this out, there was no significant change about the expression of Agr, SarA, SarX, and Rbf in ΔarcR mutant in our transcriptome results (Fold change log2 > 2 is considered significant change).
Section 3.4: The roles of Hla and Hlb in hemolysis are well defined both in terms of mode of action and regulation of their genes. The experiment involving incubation of the bacteria in the presence of blood is not defined as to what the outcome options would be and so how this would further define their roles.
Answer:
Thanks for pointing this out, we have removed the figure 4D.
Section 3.5: There is no data from an arcR icaA double deletion mutant shown in Figure 5, except for the growth curve in Figure 5C. Thus it is not possible to evaluate the claim of PIA-dependence.
Answer:
Thanks for pointing this out, we are sorry for the confusion. In fig5B, we mislabeled the notes, we have replaced ‘ΔarcR/arcR’ with ‘ΔarcR ΔicaA’. The fig 5B showed that the production of biofilm decreased after icaA was knocked out in ΔarcR mutant. As suggested, we have changed the wording about that ArcR regulates ica operon and subsequently influences biofilm.
Section 3.6 and Figure 6: The target DNA fragment sizes are listed as ica (233 bp), hla (283 bp) and hlb (251 bp), the mobility of these appear to be inconsistent with this. Were these gels electrophoresed for different times? If so, this should be indicated in the figure legend.
Answer:
Thanks for pointing this out, we have changed the images to that performed in the same conditions in figure6 to better show the results.
